


**Spatial pattern of K$_d$(PAR) and its relationship with light absorption of**
**optically active components in inland waters across China**
Zhidan Wen[a], Kaishan Song[a]*, Chong Fang[a], Qian Yang[b], Ge Liu[a], Yingxin Shang[a],
Xiaodi Wang[c]
[a]Northeast Institute of Geography and Agroecology, Chinese Academy of Sciences,
Changchun 130102, China
[b]Jilin Jianzhu University, Changchun 130118, China
[c]Harbin university, Harbin 150086, China
*Corresponding author, E-mail: songks@neigae.ac.cn
Northeast Institute of Geography and Agroecology, Chinese Academy of Sciences,
Changchun 130102, China



**Abstract**
The spatial distribution of the attenuation of photosynthetic active radiation ($K_d(PAR)$)
was routinely estimated in China lakes and reservoirs. Higher mean value of $K_d(PAR)$
was observed in Northeastern plain and mountainous region (NER). A linear model is
used to predict $K_d(PAR)$, as a function of light absorption coefficient of pigment
particulates ($a_{phy}$), colored dissolved organic matters ($a_{CDOM}$), and inorganic particulate
matters ($a_{NAP}$): $K_d(PAR) = 0.41 + 0.57 \times a_{CDOM} + 0.96 \times a_{NAP} + 0.57 \times a_{phy}$ ($R^2 = 0.87$, n=741,
$p < 0.001$). Spatial $K_d(PAR)$ was relatively dependent on the inorganic particulate
matters (average relative contribution of 57.95%). When only consider the contribution
of absorption of $a_{OACs}$ to $K_d(PAR)$, the results found that the $a_{OACs}$ could explain 70%-
87% of $K_d(PAR)$ variations. In the lakes with low TSM concnetration and non-
eutrophic lakes with high TSM, $a_{CDOM}$ was the most powerful predicting factor on
$K_d(PAR)$. In eutrophic lakes with high TSM, $a_{NAP}$ had the most significant impact on
$K_d(PAR)$. This study allowed $K_d(PAR)$ to be predicted from $a_{OACs}$ values in the inland
waters. Besides, results of this study are suggesting that new studies on the variability
of $K_d(PAR)$ in inland waters must consider the hydrodynamic conditions, trophic status
and the distribution of optically active components within the water column.
**Keywords:** light attenuation, light absorption, optically active components,
photosynthetic active radiation, inland waters





## 1. Introduction

Light is one of the most important factors governing primary production and photosynthesis in the aquatic ecosystems (Kirk, 1994; Ma & Song & Wen & Zhao & Shang & Fang & Du, 2016; Song & Ma & Wen & Fang & Shang & Zhao & Wang & Du, 2017). Light availability plays a crucial role in the distribution of phytoplankton and hydrophytes, and it is also a good indicator of the trophic state of an aquatic system. Photosynthetically active radiation (PAR) for phytoplankton growth is a product of the input of solar radiation at the surface and its reduction by optically active compounds (OACs) through absorption and scattering (Devlin & Barry & Mills & Gowen & Foden & Sivyer & Greenwood & Pearce & Tett, 2009). The diffuse attenuation of photosynthetic active radiation ($K_d(PAR)$) is commonly used to quantitatively assess the light availability, it indicates the ability of solar radiation to penetrate a water column (Kirk, 1994). $K_d(PAR)$ can be obtained by the profile of PAR values measured at different water depths according to Lambert-Beer's law (Devlin et al., 2009; Devlin & Barry & Mills & Gowen & Foden & Sivyer & Tett, 2008; Shi & Zhang & Liu & Wang & Qin, 2014). However, in situ measurements of $K_d(PAR)$ in waters have obvious limitations, and it is difficult to achieve spatial coverage. Satellite remote sensing has achieved the mapping of $K_d(PAR)$ distribution from various types of satellite remote sensing data in open sea, coastal and inland waters in recent years (Chen & Zhu & Wu & Cui & Ishizaka & Ju, 2015; Shi et al., 2014; Song et al., 2017). However, Environmental change and anthropogenic activity have made it challenging to accurately assess $K_d$ patterns in the extremely turbid inland waters (Zheng & Ren & Li & Huang & Liu & Du & Lyu, 2016). The comprehensive analysis of the relationships between $K_d(PAR)$ and $a_{OACs}$ is an imperative requirement to retrieve $K_d(PAR)$ from remote sensing data for turbid inland waters (Ma et al., 2016).





A number of components in water contribute to the attenuation of light, including
water itself, colored dissolved organic matters (CDOM), phytoplankton pigment
particles (expressed here as the concentration of chlorophyll-$a$), and inorganic
suspended particles. Water and CDOM absorb light, pigment and inorganic particles
absorb and scatter light (Effler, Schafran, and Driscoll, 1985; Shi et al., 2014).
Absorption and scattering by these OACs are the main attenuation factors of $K_d(PAR)$
in the water (Budhiman & Suhyb Salama & Vekerdy & Verhoef, 2012; Zheng et al.,
2016). The relative contribution of OACs to $K_d(PAR)$ have researched in numerous
studies previously in lakes, estuaries and offshore waters (Brandao & Brighenti &
Staehr & Barbosa & Bezerra-Neto, 2017; Lund-Hansen, 2004; Phlips, Lynch, and
Badylak, 1995; V-Balogh, Nemeth, and Voros, 2009; Yamaguchi & Katahira & Ichimi
& Tada, 2013), there is general agreement by now that inorganic suspended particles
had the decisive effect on light attenuation in turbid waters (Brandao et al., 2017; Yang
& Xie & Xing & Ni & Guo, 2005; Zhang & Zhang & Ma & Feng & Le, 2007a). In
transparent marine and freshwater systems, phytoplankton is also an important
component in PAR attenuation (Laurion & Ventura & Catalan & Psenner & Sommaruga,
2000; Lund-Hansen, 2004). Studies about $K_d(PAR)$ partition and the influencing factors
provide important information to predict the underwater light climate from the
concentrations of these factors (Brandao et al., 2017; Zhang & Zhang & Ma & Feng &
Le, 2007c). In present, the contribution of OACs to light attenuation in many studies
included both OACs absorption ($a_{OACs}$) and particulates scattering. In fact, the
particulates scattering occupied small proportion of total contribution, most of the light
attenuation in water was induced by $a_{OACs}$ (Belzile, Vincent, and Kumagai, 2002). Thus,
the relationship between $K_d(PAR)$ and $a_{OACs}$ is very essential to predict $K_d(PAR)$ based
on $a_{OACs}$ in inland waters.



Although $K_d$(PAR) characterization has been carried out in various aquatic

environments, including freshwater, estuaries, coastal water, and open ocean water
(Belzile et al., 2002; Cunningham, Ramage, and McKee, 2013; Frankovich, Rudnick,
and Fourqurean, 2017; Lund-Hansen, 2004; Zhang et al., 2007a), few studies have been
performed in the extremely turbid waters and plateau water with strong ultraviolet
radiation (Ma et al., 2016; Shi et al., 2014; Song et al., 2017). In transparent marine and
freshwater systems, phytoplankton was suggested to be an important component in light
attenuation (Brandao et al., 2017; Yang et al., 2005). However, in turbid inland waters,
the components of OACs vary independently (Matsushita & Yang & Yu & Oyama &
Yoshimura & Fukushima, 2015; Wen & Song & Zhao & Du & Ma, 2016), and studies
have pointed out that the components of OACs had large spatial and temporal variations
in turbid inland waters (Oliver & Collins & Soranno & Wagner & Stanley & Jones &
Stow & Lottig, 2017; Zhang & Zhou & Shi & Qin & Yao & Zhang, 2018; Zhao & Song
& Wen & Li & Zang & Shao & Li & Du, 2016). The governing factors controlling
$K_d$(PAR) always changed with the OACs concentration and component in different
inland waters (Brandao et al., 2017; Cunningham et al., 2013; Laurion et al., 2000).
China has a large number of inland waters, and they exhibit large variability in terms
of the optical properties and trophic status. A large proportion of lakes in China are
characterized by highly turbid waters (Song & Wen & Shang & Yang & Lyu & Liu &
Fang & Du & Zhao, 2018). Thousands of closed lakes with high salinity have developed
in the plateau area, and they are exposed to high intensity solar radiation (Laurion et al.,
2000; Ma & Yang & Duan & Jiang & Wang & Feng & Li & Kong & Xue & Wu & Li,
2011). To the best of our knowledge, there is little work has analyzed in detail the effect
of $a_{OACs}$ on $K_d$(PAR) in a large variety of inland waters across China.

In this study, our objectives were (1) to describe the spatial distribution of $K_d$(PAR)



in five limnetic regions, China; (2) evaluate which optical variables control the $K_d(PAR)$
in the water column of inland waters, especially in the different types of lakes, (3) to
provide an empirical model to estimate $K_d(PAR)$ in these inland waters. The study is
essential to remote sensing of $K_d(PAR)$ and evaluate the underwater light climate.

## 2. Materials and Methods

### 2.1. Study area and Sampling description

China is situated in eastern Asia, on the western shore of the Pacific Ocean, covering
an area of approximately $9.6 \times 10^6$ km$^2$ (E: 73°40'-135°2'30'', N: 3°52'-53°33'). China
is characterized by temperate continental climate, with a large temperature difference
between summer and winter. The spatial distribution of annual average sunshine hours
increases from southeast to northwest. There are a large number of lakes and reservoirs
with the total surface area of 104,415 km$^2$, accounting for about 1.09% of the China
total area, and this area accounts for 3.48% of the global lake and reservoir surface area
(Ma et al., 2011; Raymond & Hartmann & Lauerwald & Sobek & McDonald & Hoover
& Butman & Striegl & Mayorga & Humborg & Kortelainen & Duerr & Meybeck &
Ciais & Guth, 2013; Wen & Song & Shang & Fang & Li & Lv & Lv & Chen, 2017).
In accordance with the regions and topography, the lakes are divided into five limnetic
regions: Inner Mongolia -Xinjiang plateau region (MXR), Tibet-Qinghai Lake Region
(TQR), Northeastern plain and mountainous region (NER), Yunnan- Guizhou Plateau
region (YGR), and Eastern plain region (ER) (Wen et al., 2017). Current estimation
suggest that the total water storage of these lakes and reservoirs in China is about
1,280.75 km$^3$ (Song et al., 2018). The actual water storage of lakes in China is likely to
be greater than currently known due to underestimation of the presence of many
temporary small lakes (Song & Zang & Zhao & Li & Du & Zhang & Wang & Shao &
Guan & Liu, 2013). The trophic status of lakes in China included oligotrophic,



mesotrophic, and hypereutrophic, water quality of the majority of lakes has degraded
(Jin, Xu, and Huang, 2005).

Surveys were carried out between April 2015 and September 2017 with a total of

741 locations covered 141 lakes and reservoirs in China (here after together called
lakes). A total of 13 field surveys covering the whole country was conducted. Details
about the distribution of the sampling lakes are shown in Figure 1. These lakes
distributed in different climatic zones with various land-use types. During the sampling
period the mean day air temperatures ranged from 15 to 25 ℃. The areas of these lakes
ranged from 1 km$^2$ to 3,283 km$^2$, including freshwater and saline lakes. The surface
water (0.2-0.5 m depth) was collected in the acid-washed HDPE bottles, and were
placed in a portable refrigerator before they were carried back to the laboratory. The
location of each sampling station was recorded with a UniStrong G3 GPS. Water
samples were collected at 5-7 sampling stations from lakes on average, in the
meanwhile, PAR values were also measured in the same station. In total, PAR values
were collected in 741 stations in nine field experiments. The PAR values were
measured using the LI-COA 193SA underwater spherical quantum sensor. The
operation was conducted on the sunny side of the boat to avoid any shadow effects. The
PAR measurements were taken at no less than five point's depth for each station. At
each depth in the water, PAR value was continuously recorded for 15 s and output an
averaged value, the average value was regarded as the PAR value at this water depth
(Ma et al., 2016).





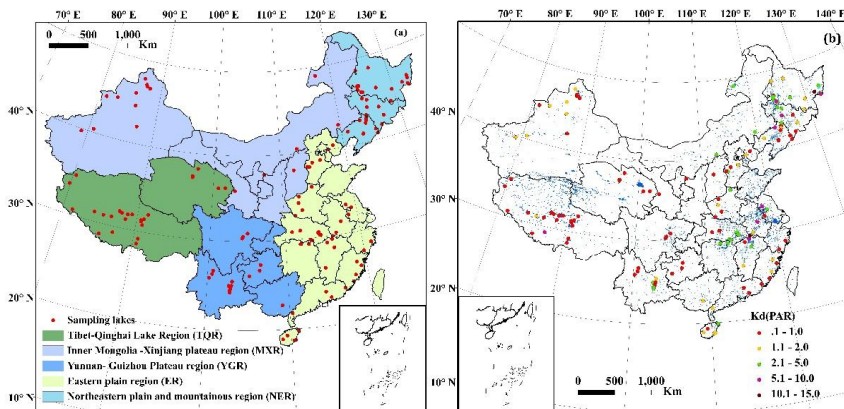

Fig. 1 Study area location and sampling lakes distribution, (a) sampling lakes distribution in five

limnetic regions, (b) $K_d$(PAR) values distribution of every sampling lake

## 2.2. Water quality and light absorption parameters measurement

Salinity and pH were measured by a portable multi-parameter water quality analyzer

(YSI 6600, U.S) in situ with the uncertainty of 0.01 ppt and 0.01, respectively. Secchi

disk depth (SDD) at each sampling site was measured using a 30 cm diameter Secchi

disk. All water samples were filtered through 0.45 μm mixed fiber millipore filters

within 24 h of sampling, and the filtered waters were used to TN concentrations analysis

by a continuous flow analyzer (SKALAR, San Plus System, the Netherlands). Total

phosphorus (TP) was determined using the molybdenum blue method after the samples

were digested with potassium peroxydisulfate (APHA, AWWA, and WEF, 1998). DOC

concentrations were also analyzed using a total organic carbon analyzer (TOC-VCPN,

Shimadzu), details can be found in the reference (Song et al., 2018). Chlorophyll a

(*Chla*) was extracted from raw water samples using a 90% buffered acetone solution,

and the concentration was determined by spectrophotometry (UV- 2600 PC, Shimadzu)

(Jeffrey & Humphrey, 1975). Total suspended matter (TSM) concentration was

determined gravimetrically, a certain volume of raw water were filtered through pre-

combusted 0.7 μm glass fiber millipore filters (Whatman, GF/F 1825-047), the



particulate matter were retained in the filters, and then the filters were combusted for
2h on 400℃. TSM concentration was calculated by the difference between filtered
combusted filter and non- filtered combusted filter(Cleveland & Weidemann, 1993).

Total particulate light absorption ($a_p$) of the filter captured TSM was determined

by UV spectrophotometry (Shimadzu, 2660) with a virgin filter as a reference
(Cleveland & Weidemann, 1993). Then the sodium hypochlorite solution was used to
remove pigments in this filter, and the bleached filter was determined again to obtain
the optical density ($OD_\lambda$) of the non-algal particles ($a_{NAP}$). The pigment or
phytoplankton light absorption coefficient ($a_{phy}$) was the difference between $a_p$ and $a_{NAP}$.
The collected water samples were filtered in turn through a GF/F 0.7 μm glass fiber
membrane and a 0.2 μm polycarbonate membrane to extract CDOM. The filtering
process should be finished within 24 h away from light. Light absorption of colored
dissolved organic matter ($a_{CDOM}$) in the waters was also measured using a UV-2600
spectrophotometer equipped a 5 cm quartz cuvette, the Milli-Q water was used as a
reference. The light absorption coefficient of CDOM at 700 nm was used to correct
CDOM absorption coefficients to eliminate the internal back scattering (Bricaud, Morel,
and Prieur, 1981). The absorption coefficients ($a_{p,}$ $a_{phy\ and}$ $a_{CDOM}$) were derived from the
measured $OD_\lambda$ *as* the following equations (Bricaud et al., 1981; Bricaud & Stramski,
1990). In this study, absorption coefficients at 440 nm was chosen for analysis later in
this study (Wen et al., 2016). The light absorption of optically active components ($a_{OACs}$)
is the sum of $a_{CDOM}$ and $a_p$. Where $a_{CDOM}$ (λ), $a_P$(λ), and $a_{phy}$(λ) are the CDOM, total
particulate and phytoplankton absorption coefficients at a given wavelength,
respectively; *L* is the cuvette path length (0.01 m); *S* is the effective area of the deposited
particle on the fiber membrane ($m^2$); *V* is the volume of the filtered water ($m^3$); 2.303
is the conversion factor; and $OD_{(null)}$ is the OD value at 700 nm.





$$a_{CDOM}(\lambda) = 2.303 \times [OD_{(\lambda)} - OD_{(null)}]/L \qquad (1)$$

$$a_P(\lambda) = 2.303 \times \frac{S}{V} \times OD_{(\lambda)} \qquad (2)$$

$$a_{phy}(\lambda) = a_P(\lambda) - a_{NAP}(\lambda) \qquad (3)$$

**2.3. Data analysis**
$K_d(PAR)$ was calculated using the exponential regression model as the following
equation, where Z is the water depth, and $PAZ_Z$ is the photosynthetic active radiation
value at depth Z (Pierson & Kratzer & Strombeck & Hakansson, 2008; Stambler, 2005).
The results were accepted only if the coefficient of determination ($R^2$) was higher than

209 0.95.

$$PAR_{Z2} = PAR_{Z1} \times e^{-K_d(PAR) \times (Z_2 - Z_1)}$$

A classification regression tree approach (CHAID) was used to classify the lakes
based on $K_d(PAR)$ in SPSS 19.0. $K_d(PAR)$ was used value as the response variable, the
explanatory variables were TSM, Chla, $a_{CDOM}$, pH, salinity, and trophic status of lakes.
Mean value and standard error of $K_d(PAR)$ were calculated for each branch of the
regression tree.
We approached data analysis in the following ways: First, the $K_d(PAR)$ differences
in different limnetic regions across China were quantified by the regional mean value
of all lakes. Meanwhile, the relative contributions of $a_{OACs}$ to $K_d(PAR)$ was calculated
according to the references (Brandao et al., 2017; Kirk, 1976; Pierson et al., 2008; Pope
& Fry, 1997). The second approach was to establish links between $K_d(PAR)$ and $a_{OACs}$
in lakes using in situ measured values of all sampling sites. Third, regression tree
analysis was used to classify the lakes based on $K_d(PAR)$ values, and the relationships
between $K_d(PAR)$ and $a_{OACs}$ in different types of lakes were explored using the
multivariate regression analysis.





## 3. Results

### 3.1 General surface water properties of lakes in different limnetic regions

In all field surveys conducted over the 141 lakes across different limnetic regions interweaved with the diverse geographical environments, a large diversity of lakes with varying water qualities was encountered. We analyzed the transparency and trophic status of these lakes, and found that lakes in the YGR had the highest transparency, followed by YGR, MXR, ER, and NER showed the lowest transparency (SDD median/mean ±standard deviation: 0.40/0.90 ±1.03 m) (Fig. 2a). The lakes in NER were highly turbid. NER is in the fluvial plains, the most of lakes in this area are shallow (2.8 ±1.8 m) with re-suspension of bottom sediments. The trophic status of lakes across different limnetic regions showed that 24.14% studied lakes in NER had a mesotrophic status, and others were all eutrophic lakes (75.86%). The proportion of eutrophication of NER lakes was the highest in five limnetic regions, followed by ER (65.67%) (Fig. 2b). Agricultural non-point pollution combined with industrial and domestic sewage discharge were the main reasons for these highly eutrophic waters in the NER and ER.

Compared with MXR, lakes in the YGR were more transparent ($1.73/2.46 \pm 2.48$ m) (Fig. 2a). It is possible that most of the lakes in the YGR are deeper tectonic ones (average: 13.8 m). Lakes in the eastern part of Inner Mongolia were shallow, and strong wind caused re-suspension, resulting in the water turbidity. In these limnetic regions, most of lakes were mesotrophic (>50%), only a few lakes were oligotrophic (<10%) (Fig. 2b). Lakes from the TQR are usually tectonic origins with a larger water depth (21.7±16.8 m), they are more transparent (3.60/4.69 ±3.62 m). Because of less human activities and limited agricultural non-point pollution, the studied lakes in this regions did not show eutrophication, over half of the sampling waters were oligotrophic (51.72%), and others were all mesotrophic status (48.48%) (Fig. 2b).





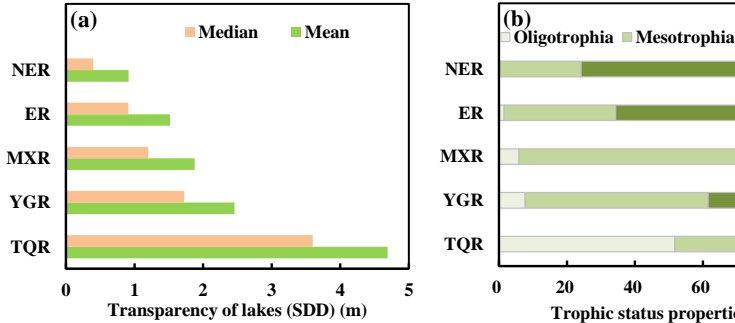

Fig. 2 Analysis of transparency and trophic status of lakes in China's five limnetic regions. (a) the

transparency analysis; (b) trophic status analysis.

### 3.2 Spatial distribution of $K_d(PAR)$

Due to the diverse geographical environments in the area of study, the sampled lakes

included the varying $K_d(PAR)$ values (Fig. 1). $K_d(PAR)$ values in different lakes ranged

from 0.11-13.93 m$^{-1}$ with the mean of 1.99 m$^{-1}$. The minimum value occurred in the

Pumoyum Co Lake of the southern Tibetan Plateau region. The maximum value

occurred in the Qingnian reservoir of Northeastern region. The average $K_d(PAR)$ value

for each of the five lake groups was calculated and ranged from 0.60 m$^{-1}$ in TQR to

3.17 m$^{-1}$ in NER (Fig. 3). In NER, the minimum value occurred in the Hengren reservoir

of 0.47 m$^{-1}$. In ER, the minimum value occurred in Haicang Lake of 0.20 m$^{-1}$. In MXR,

the minimum value occurred in Sayram Lake of 0.13 m$^{-1}$. In YGR, the minimum value

occurred in Fuxian Lake of 0.25 m$^{-1}$. In TQR, the minimum value occurred in Pumoyum

Co Lake of 0.11 m$^{-1}$.



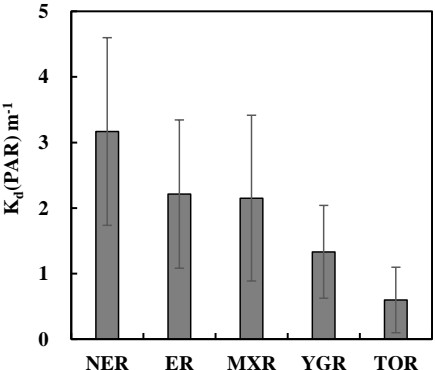


Fig. 3 Compare of mean $K_d$(PAR) values in five limnetic regions

### 3.3 Relationship between $K_d$(PAR) and OACs light absorption

A significant positive correlation was observed between $K_d$(PAR) and OACs total light
absorption in lakes across China at all sampling sites, data were evenly distributed on
both sides of the regression line (Fig. 4a). The best function to describe the relationship
through a linear model: $K_d$(PAR)$=0.86 \times a_{OACs}+0.22$ ($R^2=0.85$, n=741). The linking
between $K_d$(PAR) and light absorption of each optically active compound was also
explored. Except $a_{NAP}$ showed a significant positive correlation with $K_d$(PAR) (Fig. 4b),
they all had no significant linear relationship to $K_d$(PAR) (Fig. 4c-4d).

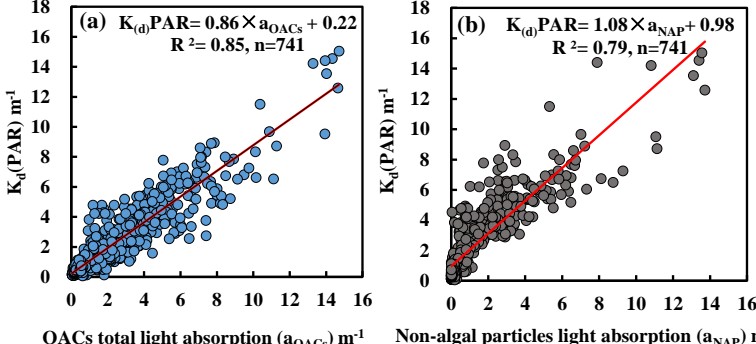






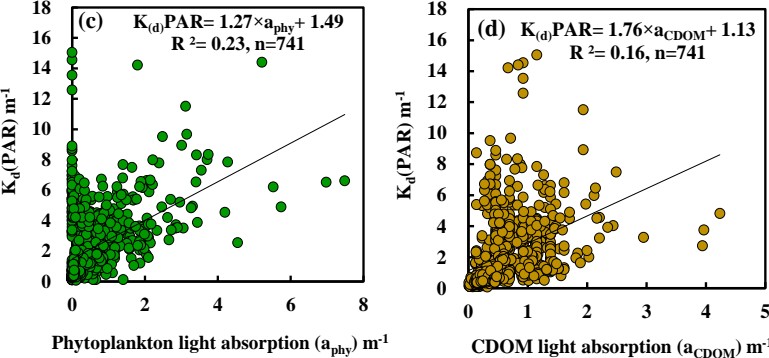


Fig. 4 Scatter plots of diffuse attenuation vs light absorption of optically active components, (a)
$a_{OACs}$, (b) $a_{NAP}$, (c) $a_{phy}$, and (d) $a_{CDOM}$

The result of multiple regression analysis showed that all the optically active

components had impact on $K_d(PAR)$, and the relational expression was as follow:
$K_d(PAR)=0.41+0.57 \times a_{CDOM}+0.96 \times a_{NAP}+0.57 \times a_{phy}$ ($R^2=0.87$, n=741, $p<0.001$) (Table
1). The standardized coefficient of independent variables indicated that $a_{NAP}$ had the
most significant impact on $K_d(PAR)$, followed by $a_{phy}$. TSM expresses the total
concentration of inorganic and pigment particulate matter in water (Budhiman et al.,
2012). The relationship between $K_d(PAR)$ and TSM was also explored to support the
regression analysis result (Fig. 5).

Table 1 Summary of multiple regression analysis

|  | R | R Square | Adjusted R Square | Std. Error of the Estimate | Sig. |
|---|---|---|---|---|---|
| All lakes | 0.931 | 0.867 | 0.866 | 0.833 | 0.000 |
| TSM <3.8 mg/L | 0.863 | 0.744 | 0.742 | 0.220 | 0.000 |
| TSM >3.8 mg/L (Non-eutrophic lakes) | 0.880 | 0.774 | 0.770 | 0.429 | 0.000 |
| TSM >3.8 mg/L (Eutrophic lakes) | 0.874 | 0.764 | 0.762 | 1.106 | 0.000 |




288 Dependent Variable: $K_d$(PAR); Predictors: constant, $a_{phy}$, $a_{NAP}$, $a_{CDOM}$

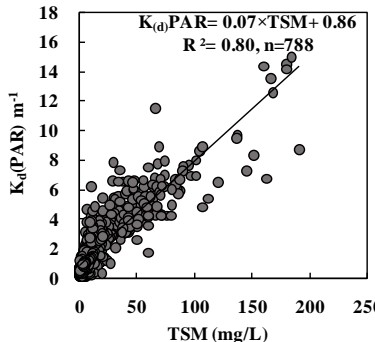


290   Fig. 5 Relationship between $K_d$(PAR) and total suspended matter concentration (TSM)

291    In five limnetic regions, the significant positive correlation was also observed

292 between $K_d$(PAR) and total light absorption of OACs (Fig. 6). The relationship

293 coefficient and fitting degree ($R^2$) all changed for lakes in different limnetic regions.

294 The regression model in TQR had the best fitting degree ($R^2 = 0.85$) and the greatest

295 relationship coefficient (slope=0.95) than in other limnetic regions. In MXR, the

296 regression model was $K_d$(PAR) $= 0.79 \times a_{OACs} + 0.08$ ($R^2 = 0.81$, n=156) with the smallest

297 relationship coefficient. In YGR, the regression model was $K_d$(PAR) $= 0.82 \times a_{OACs} + 0.33$

298 ($R^2 = 0.80$, n=156) with the lowest fitting degree.

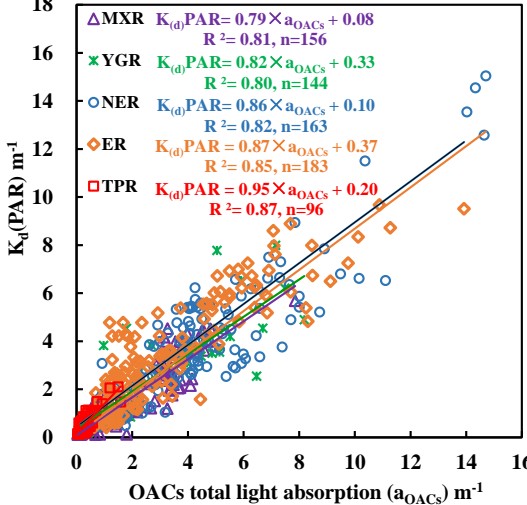




Fig. 6 Relationships between $K_d(PAR)$ and $a_{OACs}$ in five limnetic regions

In all limnetic regions in this study, $K_d(PAR)$ was dominated by inorganic
particulate matter absorption/scattering, followed by pigment particulate matters in all
limnetic regions with mean relative contributions of 57.95% and 28.20%, respectively.
The highest mean relative contribution of inorganic particulate matter to $K_d(PAR)$
(71.55 %) was in highest YGR, followed by NER (64.17 %), TQR (59.35 %), MXR
(48.26 %), and ER (46.45 %) (Fig. 7). There is a little part of the $K_d(PAR)$ variation
could be explained by CDOM with the contributions in YGR of 6.78%, in NER of
9.99%, in TQR of 10.38%, in MXR of 11.75%, and in ER of 8.71% (Fig. 7).

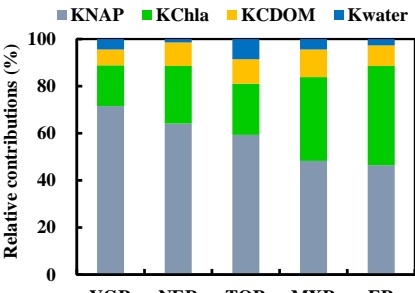


Fig. 7 Relative contributions of OACs to $K_d(PAR)$. $K_{water}$ is the partial attenuation coefficient by
pure water, $K_{CDOM}$ by CDOM, $K_{NAP}$ by inorganic suspended particles, and $K_{Chla}$ by pigment particles
**3.4 Relationship between $K_d(PAR)$ and $a_{OACs}$ in different lakes**
Regression tree analysis showed this pattern of $K_d(PAR)$ was mainly affected by TSM
concentration in these inland lakes. The $K_d(PAR)$ values in these lakes could be divided
into two branches having a TSM threshold of 3.8 mg/L. When the TSM concentration
of water was lower than 3.8 mg/L, the TSM concentration was the only predictive factor
for $K_d(PAR)$ values. However, the $K_d(PAR)$ value in lakes with the TSM concentration
higher than 3.8 mg/L, was also affected by trophic status. $K_d(PAR)$ value in oligo- and
Meso- trophic waters (mean±SD: 1.26±0.89 m$^{-1}$) was lower than in eutrophic waters





(mean±SD: 4.59±2.18 m$^{-1}$). From this point forward, the lakes are divided into two
types used 3.8 mg/L TSM concentration as a threhold: low TSM lakes and high TSM
lakes.

In order to specify the model applicability, the relationship between $K_d$(PAR) and

$a_{OACs}$ was also analyzed established for the lakes with different TSM concentration and
trophic status. The regression model for lakes with low TSM had a lower slope (slope
=0.49) than lakes with high TSM (slope =0.66, slope =0.73) with a good fitting degree
($R^2$) (Fig. 8). However, the relationship coefficient and $R^2$ all changed for lakes with
different trophic status. In the oligo- and Meso- trophic waters (non-eutrophy), the $R^2$
attained 0.70 with the relationship coefficient 0.66 (Fig. 8). In the eutrophic waters, the
regression model was $K_d$(PAR) =0.73×$a_{OACs}$+1.04 with the $R^2$ of 0.72 (Fig. 8).

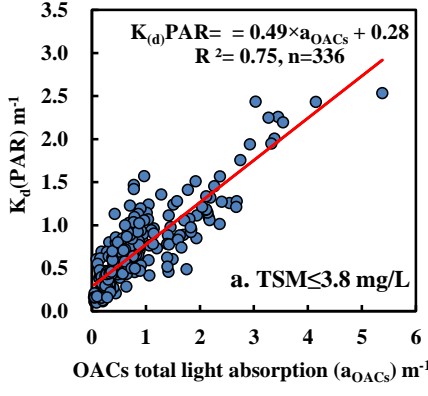


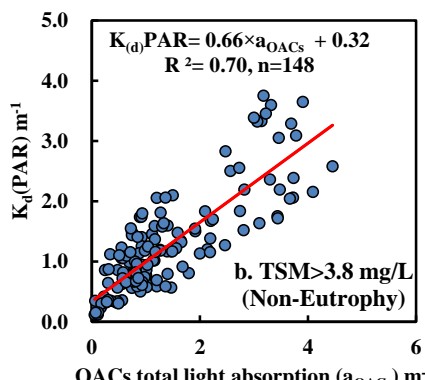




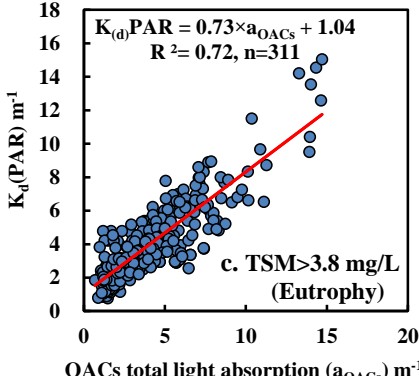


Fig. 8 Relationships between $K_d$(PAR) and $a_{OACs}$ in different lakes


In the waters with low TSM, the result of multiple regression analysis showed
$a_{CDOM}$ had the most significant impact on $K_d$(PAR), followed by $a_{NAP}$, the relational
expression was $K_d$(PAR)$=0.30+0.48\times a_{CDOM}+0.72\times a_{NAP}+0.20\times a_{phy}$ ($R^2$=0.74, $p$<0.001)
(Table 1). In the waters with high TSM, the multiple regression analysis indicated that
not all the OACs had impact on $K_d$(PAR) in oligo- and Meso- trophic waters. $a_{phy}$ was
excluded during the building of regression model. The relational expression was as
follow: $K_d$(PAR) $=0.56 +0.51 \times a_{CDOM} +0.52 \times a_{NAP}$ ($R^2$=0.77, $p$<0.001) (Table 1). The
standardized coefficient of independent variables indicated that $a_{CDOM}$ had more impact
on $K_d$(PAR) than $a_{NAP}$ in these non-eutrophic waters. In eutrophic waters with high
TSM, the regression model was $K_d$(PAR)$=1.47+ 0.35\times a_{CDOM}+ 0.82\times a_{NAP}+ 0.41\times a_{phy}$
($R^2$=0.76, $p$<0.001) (Table 1). $a_{NAP}$ had the most significant impact on $K_d$(PAR),
followed by $a_{phy}$.

## 4. Discussion


### 4.1 $K_d$(PAR) in different limnetic regions of China


In the present study, 47.37% of the in situ $K_d$(PAR) values ranged from 0.11 $m^{-1}$ to 1.00
$m^{-1}$, and 43.61% of $K_d$(PAR) ranged from 1.00 $m^{-1}$ to 5.00 $m^{-1}$, reflecting that
approximately half of these lakes are the turbid water body. The comparision of the





average $K_d(PAR)$ value in the five limnetic regions indicated that the lakes in TQR were
the most clear water, and the lakes in NER were the most turbid water (Fig. 3a). The
lake area in TQR accounts for 51.4% of total China lake area, and the majority of TQR
lakes are closed lakes with high salinity and low temperature (Ma et al., 2011; Song et
al., 2018). The lacustrine environment in TQR is suffered less interference from
anthropogenic activity with little allochthonous nutrient. The algae growth is few due
to the high salinity, low temperature, and low nutrient input, accompanying with low
Chla concentration. Moreover, the strong ultraviolet radiation in TQR could cause
CDOM photolysis and photobleaching in waters, resulting in low CDOM absorption
(Shang & Song & Wen & Lyu & Zhao & Fang & Zhang, 2018). Many large and
medium-sized lakes in TQR, developed in intermontane basin or longitudinal valley,
are the tectonic lake with deep water and steep shore. The TSM concentration in these
deep lakes may be not significantly influenced by surface runoff and wind disturbance.
According to the above reasons, the lakes in TQR may have a high water transparency,
and the attenuation of light may be relatively few than other limnetic regions. Previous
study has pointed out that most of lakes in NER were shallow lakes (Song et al., 2013),
and in shallow lakes, TSM usually plays a noticeable impact on the attenuation of light
and water transparency (Pierson, Markensten, and Strömbeck, 2003; Shi et al., 2014;
Van Duin & Blom & Los & Maffione & Zimmerman & Cerco & Dortch & Best, 2001).
TSM concentration is always higher in the shallow lakes due to the sediment re-
suspension driven by wave disturbance (Shi et al., 2014). A lake's susceptibility to
sediment re-suspension induced by wind-driven waves can be estimated by a dynamic
ratio index of 0.8 km/m (the square root of the surface area divided by the average depth)
(Bachmann, Hoyer, and Canfield, 2000). We calculated the dynamic ratios for the lakes
in NER, results showed that the values ranged from 0.82 to 10.16 km/m. All lakes in

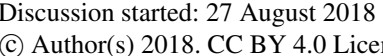



NER in this study exceeded the critical value, which supported that the resuspension
driven by winds happened in these NER lakes. The higher TSM concentration led to
the water turbidity and high $K_d$(PAR) value. These results were similar to those for
other shallow, turbid, inland waters (Shi et al., 2014; Song et al., 2017; Zheng et al.,

2016).

The $K_d$(PAR) in the water is determined by pure water and OACs, but the main

deciding factor may be different in different environments and lakes. The relative
contributions of OACs showed $K_d$(PAR) was dominated by inorganic particulate matter
absorption/scattering in all limnetic regions in this study (Fig. 7), the findings are
similar to previous findings on inland water bodies (Devlin et al., 2009; Ma et al., 2016;
Shi et al., 2014; Zhang et al., 2007a). However, there were marked regional differences
in the relative roles of inorganic particulate matter, Chl-*a* and CDOM to $K_d$(PAR) (Fig.
7). The highest relative contribution of inorganic particulate matter was presented in
YGR (Fig. 7). In this study, most of the studied lakes in the YGR are tectonic ones with
the mean deep more 10 m. The seasonal water layering is a universal phenomenon in
deep lakes (Ndebele-Murisa & Musil & Magadza & Raitt, 2014; Wetzel, 2001).
Previous studies have been demonstrated that mixing of the water column caused
resuspension of particulate matter increasing inorganic particulate matter
concentrations (James, Best, and Barko, 2004; Pierson et al., 2008; Zhang & Zhang &
Wang & Li & Feng & Zhao & Liu & Qin, 2007b), which may explain the highest
average contribution of inorganic particulate matter to $K_d$(PAR) (71.55%) in the YGR
lakes. Most of the studied lakes in YGR, over 60%, was mesotrophic with the lower
Chla concentration, except a few highly eutrophic lakes, such as Dianchi Lake and
Xingyun Lake. The algae and phytoplankton existed with an appropriate biomass, and
pigment particulate matter only had a weak contribution to $K_d$(PAR). The strong



photobleaching and photodegradation by intensive ultraviolet radiation in YGR have
destroyed CDOM structure and weakened CDOM light absorption, resulting in the
minimum contribution to $K_d$(PAR). The same phenomenon occurred in TQR (Fig. 7).
However, in ER, the relative contributions of Chla to $K_d$(PAR) is nearly equal to the
inorganic particulate matter. ER situated in the fluvial plains, and most lakes were
shallow (2.8 ±1.8 m), the waters always have high concentrations of suspended
particulate matter due to the re-suspension of bottom sediments and inflow of surface
runoff (Bachmann et al., 2000; Zhang et al., 2007c). Waters in the ER are highly turbid
with a very low transparency (0.4 ± 0.3 m). Meanwhile, the relatively high
concentrations of nutrients (TN: 0.94 ±1.31 mg/L, and TP: 0.32 ±1.02 mg/L) in lakes
resulted in phytoplankton overgrowth, even bloom. 85% of the studied lakes in the ELR
was eutrophic or hyper-eutrophic according to Carlson's trophic index (Carlson, 1977),
the pigment particulate matter during the algae decomposes and metabolism was
released to water. Many studies have proven that the controlling factor of $K_d$(PAR) was
different with variation of the region (Zheng et al., 2016). Despite Chla and CDOM
contributed to $K_d$(PAR) in ER and MXR lakes, inorganic particulate matter was largely
responsible for the attenuation. The relationships coefficient and fitting degrees ($R^2$)
between $K_d$(PAR) and $a_{OACs}$ all changed in different limnetic regions, which further
verificated indicate that the deciding factor of $K_d$(PAR) was different. This study have
indicated that althouth it sometimes had the same decisiving factor of $K_d$(PAR) in
different regions, the relative contributions of OACs to $K_d$(PAR) still had a huge
difference.
**4.2 Influence of OACs absorption on $K_d$(PAR) in lakes**
OACs have the deciding effect on $K_d$(PAR) value (Shi et al., 2014). In this study, either
in five limnetic regions or different trophic lakes, the OACs absorption and $K_d$(PAR)




had a significantly positive correlation, $a_{OACs}$ could explain 70%-87% of $K_d$(PAR)
variations (Fig 5, Fig. 8). In the whole sduty area, $a_{NAP}$ was the most significantly
regulating factor on $K_d$(PAR). The determination coefficient between $K_d$(PAR) and $a_{NAP}$
($R^2 = 0.79$) was significantly higher than that between $K_d$(PAR) and $a_{phy}$, and between
$K_d$(PAR) and $a_{CDOM}$ ($R^2=0.23$, $R^2=0.16$) (Fig. 4b-d). However, there are marked
differences in the relative contributions of $a_{OACs}$ to light attenuation in different waters
(Belzile et al., 2002; Brandao et al., 2017; Phlips & Aldridge & Schelske & Crisman,
1995; V-Balogh et al., 2009).

When the lakes were divided into different groups by TSM concnetration in this

study, the determining factor of $K_d$(PAR) changed with the lake type. In the lakes with
low TSM concnetration and non-eutrophic lakes with high TSM, $a_{CDOM}$ was the most
powerful factor on $K_d$(PAR), followed by $a_{NAP}$. The relative contribution analysis of
CDOM, Chla, and inorganic particulate matters to the total non-water light absorption
was conducted in these waters, and the results indicated that at most of these sampling
waters, CDOM absorption played a major role on total non-water light absorption, and
Chla played a minor role. These waters can be classified as "CDOM-type" water
according to the optical classification of surface waters (Prieur & Sathyendranath,
1981). Studies have indicated that in most of the highly colored inland waters, CDOM
had a dominating influence on light attenuation, reducing the amount of PAR many-
fold (Kirk, 1976; V-Balogh et al., 2009). Besides, the strong correlations between
$K_d$(PAR) and TSM also implied that light attenuation in the lakes with high TSM
concentration, the particulate absorption, including $a_{NAP}$ and $a_{phy}$, had an indispensable
influence on $K_d$(PAR) (Fig. 5). But within the PAR waveband, CDOM absorbs
maximally in the blue region of the spectrum in many natural waters (Frankovich et al.,
2017; Markager & Vincent, 2000; Morris & Zagarese & Williamson & Balseiro &



Hargreaves & Modenutti & Moeller & Queimalinos, 1995). CDOM absorption
overlaps the blue absorption maximum for Chla, which affected the light availability of
phytoplankton, resulting in the low Chla concentrtion and the low contribution of $a_{phy}$
to $K_d(PAR)$(Markager & Vincent, 2000).
In eutrophic lakes with high TSM, $a_{NAP}$ had the most significant impact on
$K_d(PAR)$, followed by $a_{phy}$. In fact, the low contribution of $a_{CDOM}$ to $K_d(PAR)$ has been
predicted since the $a_{CDOM}$ occupied a low proportion in $a_{OAC}$ (Mean $\pm$ SD: 24.30 $\pm$
14.97%) in this type of lakes. These waters can be classified as "NAP-type" water with
high TSM contrations (Mean $\pm$ SD: 40.94 $\pm$ 35.50 mg/L) and high proportion of $a_{NAP}$
to $a_{OAC}$ (Mean $\pm$ SD: 51.19 $\pm$ 22.87%) (Prieur & Sathyendranath, 1981). The
concentration of calcite particles was the most important factor regulating summer light
attenuation within Otisco Lake, New York (Weidemann & Bannister & Effler &
Johnson, 1985). In Japan Lake Biwa with bloom-forming cyanobacteria, recearchers
also found that particulate absorption played significant roles to $K_d(PAR)$ than $a_{CDOM}$
(Belzile et al., 2002). The re-suspension of bottom sediments caused by strong winds
in autumn correlated with high $K_d(PAR)$ values, which was because of the high
inorganic particles matters concentration (Ma et al., 2016; Song et al., 2017). However,
in these turbid waters, the trophic status or Chla concentration also had important
influence on light attenuation (Effler et al., 1985). Studies have pointed out that the
effect of sediments re-suspension caused by strong wind on $K_d(PAR)$ could be disturbed
by the high phytoplankton concentration in spring and summer, the algal bloom in lakes
increased the contribution of Chla to $K_d(PAR)$ (Song et al., 2017). The research on
hypertrophic waters in Hungary indicated that $a_{phy}$ played an important role in the PAR
attenuation (V-Balogh et al., 2009). Results of this study are suggesting that new studies
on the variability of $K_d(PAR)$ in inland waters must consider the hydrodynamic

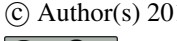



conditions, trophic status and the distribution of OACs within the waters (Brandao et
al., 2017).
The $K_d(PAR)$ in the water is governed by absorption and scattering of water,
CDOM, and particulate matter (Ma et al., 2016; Song et al., 2017; Zheng et al., 2016),
the pure water effects are always regarded as the background value of $K_d(PAR)$, so the
absorption and scattering of OACs have the deciding effect on $K_d(PAR)$ value (Shi et
al., 2014). In this study, only the contribution of OACs absorption on $K_d(PAR)$ was
analyzed and discussed. The absorption of OACs directly attenuated the photo energy
without change of light transmission direction, but the scattering of particles matters
changed light transmission direction, which resulted in the change of light absorption
along the initial transmission direction (Budhiman et al., 2012; Kirk, 1976; Zheng et al.,
2016). In fact, $a_{OACs}$ could explain most of $K_d(PAR)$ variations (Fig 5, Fig. 8), the
scattering contribution of particles matters to $K_d(PAR)$ variations in natural waters was
relatively small (Belzile et al., 2002; Lund-Hansen, 2004). The previous studies have
found that scattering of particles matters decreased approximately linearly with
increasing wavelength in particle dominated natural waters (Haltrin, 1999; Morel &
Loisel, 1998; Pegau & Zaneveld & Barnard & Maske & Alvarez-Borrego & Lara-Lara
& Cervantes-Duarte, 1999). Most of the lakes in this study had the high suspended
particles concentration, so the effect of scattering on $K_d(PAR)$ variations may be very
weak. Due to the limitation of the our experimental conditions, the scattering of
particles matters did not measured in this study, a detailed in situ profiles of spectral
absorption and attenuation measured using the AC-9 may help us to understand the
results of the research.
**5. Conclusions**
The spatial distribution of average $K_d(PAR)$ in five limnetic regions China showed that



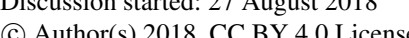


the minimum value in TQR (0.60±0.99 and the maximum in NER (3.17±2.86 m$^{-1}$). The
inorganic particulate matters had the highest average relative contribution to $K_d$(PAR)

(57.95%).

The $a_{OACs}$ could explain 70%-87% of $K_d$(PAR) variations with the following

relationship: $K_d$(PAR)= 0.41+ 0.57×$a_{CDOM}$+ 0.96×$a_{NAP}$+ 0.57×$a_{phy}$ ($R^2$=0.87, n=741,
$p$<0.001). However, the influence of different components of $a_{OACs}$ on $K_d$(PAR)
changed with the lake type. In the lakes with low TSM concnetration and non-eutrophic
lakes with high TSM, $a_{CDOM}$ was the most powerful factor on $K_d$(PAR). In eutrophic
lakes with high TSM, $a_{NAP}$ had the most significant impact on $K_d$(PAR), followed by
$a_{phy}$. A precise understanding the effect of OACs absorption on $K_d$(PAR) is essential to
remote sensing of water color and evaluate the underwater light climate.

## Acknowledgments

This study was jointly supported by the National Natural Science Foundation of China
(No. 41730104, No. 41701423), the "One Hundred Talents Program" of the Chinese
Academy of Sciences granted to Kaishan Song, 13th Five-Year Plan of Technical and
Social Research Project for Jilin Colleges (JJKH20170257KJ), and Jilin Scientific &
Technological Development Program (No. 20160520075JH).

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
