# Peer review of "Spatial pattern of K$_d$(PAR) and its relationship with light absorption of optically active components in inland waters across China"

_Biogeosciences, 2018_

## Referee Comment (RC1) · Anonymous Referee #1 · 12 Oct 2018

General comments: Underwater life is markedly influenced by the light field in the water. The spectral composition of light, its total amount, and change with depth are determined by the solar irradiance entering into water as well as by the optical properties of the natural water. Therefore, lake waters can be classified based on their optical properties and the classes indicate certain relationships related to ecological processes in these waters. The research topic is relevant because it enables better understanding optical properties in lakes as well as enhance the development of management strategies to restore and improve the ecological status of lakes In this manuscript, authors describe a new approach to predict Kd(PAR) in turbid inland waters using the absorption characteristics of optically-active components (OACs) in waters. OACs information

can be retrieved from widely available satellite images, thus allowing large-scale and high frequency assessment of photosynthetic active radiation and ecological health of lakes. To demonstrate the new approach, they used data collected from 141 lakes and reservoirs over a 3-year period. The study rationale and objectives are well stated and grounded in existing literature. Methodology is sound and adequately described, and conclusions of the study are supported by the data presented. The manuscript is publishable, but the text requires a great deal of editing. My detailed comments are listed below.

The abstract doesn't include all important from the paper: for example, the aim of the study was not even mentioned. The abstract should be more concrete too! "This study highlights the . . ."; ". . . findings which have application for monitoring Kd(PAR)..." In manuscript, why are you going from OACs to aOAC? Sometimes, you write OAC, other time you write aOACs. Unless there is a valid reason, it is important to use the same abbreviation throughout the manuscript. Line 38 and throughout the manuscript, the citations should be presented as the followed format (First author, et al., year), please do not list all authors in the text. Line 43 "optically active compounds" and "optically active components". You need to develop some consistency regarding the use of terms. Line 61-64 A reference is required here. Line 117-137 Condense these sentences. Limit your description to details that would help readers under the context of the study. This section is not your emphasis. You can reduce the length as soon as possible. Line 138-144 Same thing here. These details are superfluous. Please do not in so much detail. Line 160 I care how long the samples are stored in the field and how to be stored. Line 211 Has this classification been used before? If so, provide reference. In addition, and more importantly, this is the only time reference is made to CHAID approach in the entire manuscript. There is no reason to list the method, if it is not used in the analysis of data presented in the manuscript. Line 221 I think, it's more precise to say "point". An entire lake can be viewed as a study site. Line 227 That sentence is a repetition of information that has just been provided. Line 234 Sometimes you use trophic status but sometimes trophic state. Please the consistent

[Figure]

expression for all the text. Line 314 What is the basis for using a TSM threshold of 3.8 mg/L to categorize the lakes? If that is based on a previous study, list the reference(s). If that is based on the analysis of the data from the present study, indicate where the information is presented. Figure 2 and Figure 8: I could not find in the text how the trophic states were defined.

---

## Author Comment (AC1) · 19 Oct 2018

General comments: Underwater life is markedly influenced by the light field in the water. The spectral composition of light, its total amount, and change with depth are determined by the solar irradiance entering into water as well as by the optical properties of the natural water. Therefore, lake waters can be classified based on their optical properties and the classes indicate certain relationships related to ecological processes in these waters. The research topic is relevant because it enables better understanding optical properties in lakes as well as enhance the development of management strategies to restore and improve the ecological status of lakes In this

manuscript, authors describe a new approach to predict Kd(PAR) in turbid inland waters using the absorption characteristics of optically-active components (OACs) in waters. OACs information can be retrieved from widely available satellite images, thus allowing large-scale and high frequency assessment of photosynthetic active radiation and ecological health of lakes. To demonstrate the new approach, they used data collected from 141 lakes and reservoirs over a 3-year period. The study rationale and objectives are well stated and grounded in existing literature. Methodology is sound and adequately described, and conclusions of the study are supported by the data presented. The manuscript is publishable, but the text requires a great deal of editing. My detailed comments are listed below. Response: We thank the reviewer for the very instructive and helpful suggestions for revision. We have revised manuscript according to the reviewer's suggestion, and the details are listed as following. Thank you very much again for the positive evaluation and giving us the chance to further improve our manuscript. My detailed response to the comments are listed in the PDF file.

Please also note the supplement to this comment:
https://www.biogeosciences-discuss.net/bg-2018-348/bg-2018-348-AC1-supplement.zip

―――――――――――――――――――

---

## Author Comment (AC2) · 19 Oct 2018

We thank the reviewer for the very instructive and helpful suggestions for revision. We have revised manuscript according to the reviewer's suggestion, and the details are listed as following. Thank you very much again for the positive evaluation and giving us the chance to further improve our manuscript.

Please also note the supplement to this comment:
https://www.biogeosciences-discuss.net/bg-2018-348/bg-2018-348-AC2-supplement.zip

---

## Referee Comment (RC2) · R. Gonçalves-Araujo (Referee) · 24 Oct 2018

Light is an important factor controlling primary production in the water and its measurements have been extensively applied for monitoring water quality in the past centuries. Significant technologic advances have been reached in the past decades with respect to applications for monitoring light through ocean color remote sensing and autonomous in situ sensing. That allowed for a more comprehensive and high spatial- and temporally resolved monitoring of water quality in aquatic environments. The light attenuation, expressed through the diffuse light attenuation coefficient (Kd), controls the vertical distribution of plants and phytoplankton over the water column. It is mainly

governed by the attenuation by the water itself and the concentrations of inherent optical properties (IOPs) of e.g., phytoplankton, inorganic particles and chromophoric dissolved molecules, what makes it an important parameter for eutrophication monitoring. The study by Zhidan Wen and colleagues brings a large dataset from a consistent number of lakes sampled in China. The research topic is relevant and the dataset is of great importance for the bio-optical community as providing a tool for inland water-bodies monitoring. The proposed objectives are reasonable and provide insights into the spatial variability and determining factors of light attenuation over China lakes and reservoirs as well as a model for retrieving Kd(PAR) from measurements of optically active constituents of water. Although the objectives are sound and can be reached with the dataset used in the study, the manuscript lacks some information and data analysis and it has many rooms for improvement. Therefore, I judge that this current version of the MS is not acceptable for publication in BG and requires extensive reviews prior to re-submission.

GENERAL COMMENTS: - I understand that English might not be the primary language for the authors (the same applies for me), but the manuscript has several grammar and spelling errors and must be revised by a native speaker; - It seems that there is a misuse/misinterpretation of some concepts and terminologies along the manuscript. Some examples on odd use of terminology: phytoplankton pigment particulates, particulate matters, dissolved organic matters. Additionally, authors keep changing the terminology through the MS and that makes it confusing and difficult to follow. There must be a consistency with the terminology adopted throughout the text, and the terminology is expected to follow an updated nomenclature. - Awkward referencing style (e.g. Line 50) and too many old references. I have also found several mistakes in the reference list and citations through the MS, authors must make a thorough review on that. - Abstract: authors do not state their objectives and can be more concise and informative. - Introduction: it is a bit too long and there are some repetitions along the section. Authors can make it more concrete. - Methods: the methodology is not explained in full detail. To my understanding, the major problem of this section is that it does not provide

enough information for the reader to understand the sampling strategy adopted for this study. Authors say that the lakes vary a lot in size, but do not mention how such a variability influenced, for instance, the decision on the amount of sampling points per lake, etc. Additionally, it is mentioned that 13 field surveys were performed within bit more than two years period. How was the seasonal coverage of those campaigns? Were all the 141 lakes sampled in all of those campaigns? How was the vertical sampling strategy for OACs? How was the bottom depth variability within the lakes and how did the authors deal with that? - Results: are generally well-presented, however, some parts are confusing and difficult to follow. One thing is missing here. . . The authors propose a model for estimating Kd(PAR) from OACs, but they did not perform any validation exercise. I sincerely expected to see at least a scatter plot with the Kd(PAR) measured in situ against the Kd(PAR) obtained with the model presented here. Additionally, results presented in the manuscript could be shortened. Authors present figures with the equations for obtaining Kd(PAR) from each of the OACs individually, then with all together and finally an equation taking into account all of the OCAs (as an individual term of the equation). This can be simplified by only presenting the equation from line 337, which is the most important result here. Since kd is a function of all the OCAs it does not make sense to show graphs for all the OACs individually. - Discussion: this section is weak. Authors are expected to discuss their results in deep, explain their findings and support them with the literature. However, in this piece of work it is observed rather some comparisons with other studies or only some random sentences scattered along the section, that do not help or add anything to the discussion itself. - Conclusion: the section is short, and it only presents a wrap up of the results instead of a conclusion that highlights the overall importance of the main findings of the study.

SPECIFIC COMMENTS: L 20-22: "absorption coefficient of pigment particulates", "dissolved organic matters" and "inorganic particulate matters" seems a bit odd. I suggest the authors to adopt "absorption coefficient of phytoplankton, colored dissolved organic matter and inorganic particles". L 24-25: Need to clarify whether it is considering each of the OACs individually or it takes into account the sum of OACs (i.e., total non-water

absorption – atw). L 30-32: I cannot see how your results support the affirmation. L 42-45: PAR is also attenuated by the water itself. L 51-52: not clear what it meant with that sentence. L 54-58: How do environmental change (what do you mean by that?) and anthropogenic activity make it challenging to assess Kd in turbid inland waters? L 58-60: Not only for inland waters, it is actually required to all aquatic environments. L 62-63: "phytoplankton pigment particles (expressed here as the concentration of chlorophyll-a)" – the concentration of Chl-a is an index for the phytoplankton biomass. Absorption by phytoplankton is represented as the absorption coefficient of phytoplankton. L 68-74: authors may want to rephrase this sentence. L 78: "underwater light climate" – I suggest the authors to replace by "underwater light field" L 80-83: those sentences do not make sense to me. Authors may want to rephrase them. L 83-85: it applies for all aquatic systems. L 90: what do the authors mean by plateau waters? And why do they receive such a strong UV radiation? L 91-93: repetition of lines 74-77. Additionally, there is no reference for marine studies. L93-94: this condition is not unique for turbid inland waters. L 96: what do you mean by large spatial variability? Is that intra- or inter waterbodies? L 100: what do the authors mean by "OACs component"? L 105: give salinity intervals. How much is salinity for a lake? L 110: Objective 1 – it is not clear, whether the authors want to compare variability among the regions or if they want to assess the spatial variability in each of the regions. L 111: what do you mean by "optical variables"? Maybe OACs? L 112: what do you mean by "especially in the different types of lakes"? L 113: Objective 3 – provide the model based on what? L 120-121: Awkward phrasing. L 127: What do you mean by "in accordance with the regions and topography"? Are those socio-economic regions? Geomorphologic/Climate regions? Additionally, I did not check for it, but I suspect that there might be other factors other than "regions and topography" behind the division of those regions. Looks a bit simplistic. It is not clear along the MS what kind of regions are those. Authors need to make it clear. L 134: "temporary small lakes" – do you mean perennial ? L 135: Oligotrophic lakes: you use this terminology here and another few times along the MS and then changes to non-eutrophic lakes. It is necessary to have consistency when

it comes to terminology. L 135-136: this sentence does not make sense. Consider rephrasing it. L 138-141: In what seasons were the surveys carried? All year round? This has great influence on solar radiation. How many stations were performed per lake? Where were the stations located in the lakes? In the borders, in the center? Authors may want to provide such information regarding sampling sites, stations performed, sampling depths, etc. as supplement file to this MS. L 142-143: This sentence is confusing and could be easily removed from the MS. L 144: since there is a great variability in the area of sampling lakes, how was the strategy regarding the number of sampling stations with respect to the area of the lakes? - How were the water samples collected? There is no information on that. Were there only surface samples collected? How confident are the authors by providing a model for light attenuation over the water column based on surface measurements? L 149-156: It is not 100% clear how the PAR measurements were conducted. Was there a surface reference measurement? What was the general vertical resolution of PAR measurements? Was the PAR measurements spectrally resolved, or was provided an averaged PAR value? If spectrally resolved, what are the channels? L 170-172: Not clear whether it was measured directly in water samples or in filters containing the cells. L 179-181: please provide more information on the equipment. Spectral resolution and range? Was it measured with an integrating sphere? L 187-195: what was the spectral resolution and range of the measurements? L 194: why was the 440nm wavelength chosen? L 198: Would not "L" be 0.05? L 198-199: How was the effective area of the deposited particles on the filter was measured? L 205-210: Please provide more information on the calculations of Kd. How was it obtained? Calculations were made for each wavelength and then averaged (spectrally resolved) or PAR was averaged per depth? L 208: r2 was obtained based on the relationship of which pair of variables/parameters? L 212: What is SPSS 19.0 ? L 213: How was the trophic status of lakes assessed in this study? L 229: what do you mean by transparency and how was it measured in this study? L 229-230: Again, how was the trophic status assessed? L 231: Please clarify how the transparency was measured. Is that Secchi disc depth? L 234: Missing references.

L 234-236: Again, how was it measured? What are these results about? What do those numbers mean? L 237: What do you mean by "proportion of eutrophication"? L 238-239: Please, reference accordingly. L 245: "tectonic origins" – Reference for that? L 246-249: Looks like discussion. Question: How was the correlation between transparency and trophic state? And kd vs transparency? I suspect there might be a significant correlation between these variables. L 254-255: Sentence needs review. L 257-258: Stick to the abbreviations provided. It is confusing when alternating between calling the regions by name and abbreviation. L 260-264: Confusing and not very informative. Most of the readers have no idea about the locations of the referred lakes. Could be removed from the text. Perhaps authors may want to devote a bit more of effort to describe lakes that have high social-economic impact in the country, instead. L 269: "at all sampling sites" – what do you mean by that? Was the correlation tested for each site individually? It is not clear. L 271: What that the best linear regression, or was a linear regression the best one to describe the correlation? Have you tested for other types of functions? L 279-280: "all the optically active components had impact on Kd(PAR)" – what do you mean by that? L: 282: what is the standardized coefficient of independent variables? Where are those results presented/discussed in the MS? L 285: The correlation between Kd and TSM was greater than for kd vs aNAP, how do the authors explain that, since they say that aNAP has the most significant impact on kd(PAR)? L 294-295: Any reason/hypothesis why that region had the best performance for predicting kd(PAR)? L 303: those relative contributions are related to what? L 304-308: confusing sentences. The authors may want to rephrase them. L 315: How was the 3.8 mg/L threshold defined? What is the reason for that value? L 318-319: why to combine oligo- and mesotrophic lakes? L 320-322: What about the limnetic regions? Why to use this classification? Any reference for that? Is there any clustering for those stations? In figure 8 the TSM threshold division is further subdivided into non-eutrophy and eutrophy. Authors should state the reason for performing such divisions. L 328-329: Confusing sentence. L 336: what do you mean by "relational expression"? L 338-339: where are those results shown? L 339-340: why was aphy excluded? Authors have to estate the reason for that. Results – suggestion: I have the impression that the authors wanted to include all their results in the MS. However, some of those results could be omitted without changing the concept of the manuscript and it would make it easier to present, write and follow. Authors should rethink what results are worth it to be presented in the manuscript. L 349-351: Based on what could you infer that? Are there any thresholds? Please, provide reference. L 357-359: Please provide a reference for that statement. L 361-363: please provide reference. L 363-365: please provide reference. L 371-372: Do you mean wind-induced waves? What about the establishment of the seasonal thermocline? What about the allochthnous input of TSM? What can you say about it? L 379-381: The sentence does not add information to the discussion. Consider removing it or developing it more in deep. L 386: where are those similar inland bodies? Please develop more the discussion instead of making comparisons. L 401-404: CDOM photobleaching and photodegradation: how can the authors infer that based on their results? If the information is from other study, please, include the reference. L 410-412: How can the authors infer that form their results? Have you measured phytoplankton biomass prior to the "overbloom"? L 41.415: Not clear what the authors want to say in that sentence. L 415-416: Many studies and only one reference? L 425: Awkward phrasing. L 428: Figure 5 only shows TSM results. L 428-429: What was the overall absorption budget for the studies regions? I think that your results would be clearer if you present a ternary plot for the absorption of OACs (CDOM x phytoplankton x NAP) for each of the regions. L 431-432: Not clear what the authors want to say in that sentence. L 439: "Chla" – is not it the contribution of phytoplankton, which is expressed by means of Chla concentration? L 438-442: where are those results presented in the manuscript? L 442-444: that classification was made for oceanic waters. Additionally, the terminology presented in the manuscript is out of date. L 444-449: what can the authors discuss/add/conclude about that information? How does it help the interpretation of their results? L 449.455: those sentences can be deleted without changing the interpretation of results and discussion of this manuscript. L 461-463: This sentence does not add to the discussion given that the authors do not

mention calcite particles in their study. L 469: are you assuming Chla as a proxy for trophic status? L 470-476: Not clear what the authors want to say in that sentence. Additionally, the authors start the sentence with the word "Studies" and present only a single reference. L 475-478: how do the results suggest that? It is not clear to me. L 484-487: Where is this shown along the MS? How do your results support such an affirmation? L 488-490: How do your results support that affirmation? How was the scattering contribution in this study? L 490-496: Not clear how those sentences would help/add to the discussion. L 498: I would suggest an AC-S instead, given the much better spectral resolution. L 501: The sentence does not make sense. L 506: when presenting the given, please indicate at what wavelength the absorption coefficients should be considered. L 511-512: Sentence does not add to the conclusions. I suggest to remove it.

FIGURES AND TABLES: Captions provide a poor description of figure contents. Authors should put more effort on that. Figure in general are well presented and I have some specific comments/suggestions below: Figure 1. Very poor resolution and it is almost impossible to read the text and see the inset figure in each panel. In (a) there is no reference for the limnetic region definitions. What do the red dots mean? What is presented in the inset graph? Additionally, the kd(PAR) values presented in panel (b) are not described. Where was that data from? Figure 2: Provide more information in the caption. Figure 3: Have you tested for the differences among regions? It seems that there is no significant difference between ER and MXR. Figure 4. Please indicate the selected wavelength for absorption coefficients. It is mandatory to provide such an information. Table 1. What do you mean by adjusted r2? How did you get to that? Figure 5. How is that possible to have a n=788, when you mention that only 741 samples were taken? Figure 7. How was the K values for each OACs obtained? How was the Kwater obtained? I suggest you to make ternary plots instead. It gives more information on the absorption budget and you can split it into different seasons/years to see how it varies over time. Finally, given that the authors present a model to retrieve Kd(PAR), I expected to see a figure where calculated Kd was plotted against in situ

observed Kd.

---

## Author Comment (AC3) · 6 Dec 2018

**Response to reviewer**

**Title: Interactive comment on "Spatial pattern of Kd(PAR) and its relationship with light absorption of optically active components in inland waters across China**

**Referee: Rafael Gon calves-Araujo**

SPECIFIC COMMENTS:

1. L 20-22: "absorption coefficient of pigment particulates", "dissolved organic matters" and "inorganic particulate matters" seems a bit odd. I suggest the authors to adopt "absorption coefficient of phytoplankton, colored dissolved organic matter and inorganic particles".

Response: Thank you for the suggestion, and we have adopt the suggestion in the revised manuscript. 2. L 24-25: Need to clarify whether it is considering each of the OACs individually or it takes into account the sum of OACs (i.e., total non-water absorption – atw).

Response: It takes into account the sum of OACs, we used "the total non-water absorption" in the revised manuscript.

3. L 30-32: I cannot see how your results support the affirmation.

Response: This study analyzed the spatial distribution of the  $K_d(PAR)$  in China lakes and reservoirs. the relative contribution of CDOM, Chla, and inorganic particles to the total non-water light absorption, and the results showed that when only consider the contribution of absorption of  $a_{OACs}$ to  $K_d(PAR)$ , the total non-water absorption could explain 70%-87% of  $K_d(PAR)$  variations (Figure 6). In the lakes with low TSM concnetration and non-eutrophic lakes with high TSM,  $a_{CDOM}$  was the most powerful predicting factor on  $K_d(PAR)$ . In eutrophic lakes with high TSM,  $a_{NAP}$  had the most significant impact on  $K_d(PAR)$  (Figure 8). These results can support the affirmation in line 30-32.

4. L 42-45: PAR is also attenuated by the water itself.

Response: This is very correct, and we have added this in the revised manuscript.

5. L 51-52: not clear what it meant with that sentence.

Response: We have re-written the sentence. "However, this traditional measuring approach in situ is not well suited for assessment of  $K_d(PAR)$  at large spatial scale."

6. L 54-58: How do environmental change (what do you mean by that?) and anthropogenic activity make it challenging to assess  $K_d$  in turbid inland waters?

Response: Dramatic environmental changes have taken place in many inland lakes. For example, a decrease in lake area has resulted from lake reclamation. Anthropogenic activities like dam construction have also readjusted the hydrological conditions. Owing to changes in water quantity and sediment discharge, the environmental change and anthropogenic activity make it challenging to assess  $K_d$  in turbid inland waters.

Zhang, G., Xie, H., Yao, T., Kang, S.: Water balance estimates of ten greatest lakes in China using ICESat and Landsat data, Chin. Sci. Bull., 58(31), 3815-3829, 2013.

7. L 58-60: Not only for inland waters, it is actually required to all aquatic environments. L 62-63:

"phytoplankton pigment particles (expressed here as the concentration of chlorophyll-a)" – the concentration of Chl-a is an index for the phytoplankton biomass. Absorption by phytoplankton is represented as the absorption coefficient of phytoplankton. L 68-74: authors may want to rephrase this sentence. L 78: "underwater light climate" – I suggest the authors to replace by "underwater light field" L 80-83: those sentences do not make sense to me. Authors may want to rephrase them. L 83-85: it applies for all aquatic systems. L 90: what do the authors mean by plateau waters? And why do they receive such a strong UV radiation?

Response: We totally agree with you, and have re-written these sentences according to your suggestion.

8. L 91-93: repetition of lines 74- 77. Additionally, there is no reference for marine studies. Response: We have deleted the sentence in line 91-93.

9. L93-94: this condition is not unique for turbid inland waters. L 96: what do you mean by large spatial variability? Is that intra- or inter waterbodies?

Response: The components of OACs had large spatial variations in different turbid inland waters.

10. L 100: what do the authors mean by "OACs component"?

Response: OACs component includess phytoplankton, colored dissolved organic matter and inorganic particles.

11. L 105: give salinity intervals. How much is salinity for a lake?

Response: We have added the range of salinity (1-43.44 psu) for the lakes.

12. L 110: Objective 1 - it is not clear, whether the authors want to compare variability among the regions or if they want to assess the spatial variability in each of the regions.

Response: Objective 1 is to compare the mean Kd(PAR) values in five limnetic regions

13. L 111: what do you mean by "optical variables"? Maybe OACs?

Response: "optical variables" in here mean OACs.

14. L 112: what do you mean by "especially in the different types of lakes"?

Response: The different types of lakes means fresh water lakes and saline lakes, lakes with different TSM concentrations, lakes with different trophic status.

15. L 113: Objective 3 – provide the model based on what?

Response: The model is provided based on the relationship between Kd(PAR) and aOACs.

16. L 120-121: Awkward phrasing.

Response: We have deleted the sentence.

17. L 127: What do you mean by "in accordance with the regions and topography"? Are those socioeconomic regions? Geomorphologic/Climate regions? Additionally, I did not check for it, but I suspect that there might be other factors other than "regions and topography" behind the division of those regions. Looks a bit simplistic. It is not clear along the MS what kind of regions are those. Authors need to make it clear.

Response: Thank you for the suggestion. This division of Chinese lakes is generally accepted in China, and published in the book of Wang & Dou in 1998. We have added the information about

the partition principle. Except the regions and topography, the differences in the climatic and geographical conditions around lakes are also considered.

Wang, S. M., Dou, H. S. 1998. Record of Chinese Lakes. Science Press. 18. L 134: "temporary small lakes" – do you mean perennial?

Response: The "temporary small lakes" mean that some small lakes are seasonal presence. In the dry season, these lakes may are likely to dry up, and in the monsoon, these lakes could appear.

19. L 135: Oligotrophic lakes: you use this terminology here and another few times along the MS and then changes to non-eutrophic lakes. It is necessary to have consistency when it comes to terminology.

**Response:**

20. L 135-136: this sentence does not make sense. Consider rephrasing it.

Response: Thank you for the suggestion, and we would keep the consistency in the revised manuscript.

21. L 138-141: In what seasons were the surveys carried? All year round? This has great influence on solar radiation. How many stations were performed per lake? Where were the stations located in the lakes? In the borders, in the center? Authors may want to provide such information regarding sampling sites, stations performed, sampling depths, etc. as supplement file to this MS.

L 144: since there is a great variability in the area of sampling lakes, how was the strategy regarding the number of sampling stations with respect to the area of the lakes? - How were the water samples collected? There is no information on that. Were there only surface samples collected? How confident are the authors by providing a model for light attenuation over the water column based on surface measurements?

Response: The surveys were all carried in April and September. Water samples were collected at 4-7 sampling points from lakes on average, these sampling points were evenly distributed across the lake. The number of sampling stations in lake were linking to the area of the lake, it always had 4 points in the lake with area less than 10 km2, 5-6 points in the lake with area of 10-1000 km2, and 7 points in the lake with area over 1000 km2. The surface water (0.2-0.5 m depth) was collected by the portable water collector. The authors provided a model for light attenuation, this model was not only based on surface measurements, the PAR measurements were conducted over the water column. 22. L 142-143: This sentence is confusing and could be easily removed from the MS.

Response: We have deleted the sentence from the MS.

23. L 149-156: It is not 100% clear how the PAR measurements were conducted. Was there a surface reference measurement? What was the general vertical resolution of PAR measurements? Was the PAR measurements spectrally resolved, or was provided an averaged PAR value? If spectrally resolved, what are the channels?

Response: Thank you for the question, and the PAR measurement was first conducted at the water surface, this depth just inundated the spherical quantum sensor of LI-COA 193SA. Then the sensor was vertically dropped down in the water until the PAR values was less than 1% of the PAR value in the water surface. In this dropping down process, the PAR measurements were taken at no less

than five point's depth for each station. At each depth in the water, PAR value was continuously recorded for 15 s and automatically output an averaged PAR value, the average value was regarded as the PAR value at this water depth.

24. L 170-172: Not clear whether it was measured directly in water samples or in filters containing the cells.

Response: The water samples with the fixed volume were filtered through 0.45  $\mu$ m mixed fiber millipore filters (Bandao Industrial Co., Ltd, China) within 24 h of sampling, and these filters were used for Chlorophyll a (*Chla*) extracted using a 90% buffered acetone solution, the *Chla* concentration in the extract was determined by spectrophotometry (UV- 2600 PC, Shimadzu).

25. L 179-181: please provide more information on the equipment. Spectral resolution and range? Was it measured with an integrating sphere? L 187-195: what was the spectral resolution and range of the measurements?

Response: Light absorption of colored dissolved organic matter ( $a_{CDOM}$ ) in the waters was measured using a UV-2600 spectrophotometer equipped a 1 cm quartz cuvette, the scan range was 200-800 nm, the Milli-Q water was used as a reference. The spectral range of UV-2600 spectrophotometer is 185-1400 nm, and the spectral resolution is 1 nm.

26. L 194: why was the 440 nm wavelength chosen?

Response: We have added the reference in here. The principle and highest phytoplankton pigments absorption is located at 443 nm. Therefore the effect of phytoplankton absorption on total absorption is highest here. The CDOM absorption in visible range have overlaps with phytoplankton pigments absorption at 443, and this effect was introducing errors in ocean color remote sensing algorithms for retrieval of chlorophyll a concentration. In most cases chlorophyll a was overestimated by those algorithms that were not taking into account CDOM absorption at 443 nm. That was a reason for reporting  $a_{CDOM}(443)$  or  $a_{CDOM}(440)$  in literature, and inclusion of this parameter particularly in semi-anlytical remote sensing algorithms. Please also check the added article listed below.

- Prieur, L., Sathyendranath, S.: An optical classification of coastal and oceanic waters based on the specific spectral absorption curves of phytoplankton pigments, dissolved organic matter, and other particulate materials, Limnol. Oceanogr., 26(4), 671-689, 1981.
- Lee, Z. P., Carder, K. L., Arnone, R. A.: Deriving inherent optical properties from water color: a multiband quasi-analytical algorithm for optically deep waters, Appl. Opt., 41(27), 5755-5772, 2002.

27. L 198: Would not "L" be 0.05?

Response: Because "L" is the cuvette path length, in this study, the cuvette with diameter 0.01 m was used during the analysis process.

28. L 198-199: How was the effective area of the deposited particles on the filter was measured? Response: the effective area is the area of filter covered by particles.

29. L 205-210: Please provide more information on the calculations of Kd. How was it obtained? Calculations were made for each wavelength and then averaged (spectrally resolved) or PAR was

averaged per depth? L 208:  $r^2$  was obtained based on the relationship of which pair of variables/parameters?

Response: PAR values were measured no less than five point's depth in every sampling point using the LI-COA 193SA. At each depth in the water, PAR value was gotted as an averaged value. So in one sampling point, there are at least five PAR value with different depth. Then, in every sampling point,  $K_d(PAR)$  was determined by applying the exponential regression model which utilizes Equation (4), this regression process was based on the PAR values in no less than five point's depth.

$$PAR_{Z2} = PAR_{Z1} \times e^{-K_d(PAR) \times (Z_2 - Z_1)}$$
(4)

From the regression model (4), we could infer that Kd(PAR) was an slope value.

$$K_d(PAR) \times (Z_1 - Z_2) = ln(PAR_{Z2}/PAR_{Z1})$$

where Z is the water depth, and  $PAZ_Z$  is the PAR value at depth Z. The results were accepted only if the coefficient of determination ( $R^2$ ) was higher than 0.95.

30. L 212: What is SPSS 19.0?

Response: This is a data analysis softsare. The total name is SPSS statistics 19.0.

31. L 213: How was the trophic status of lakes assessed in this study? L 229-230: Again, how was the trophic status assessed? L 234-236: Again, how was it measured? What are these results about? What do those numbers mean?

Response: We have added this method in the manuscript. The assessment of the trophic status of lakes was based on the modified Carlson's trophic state index (TSI) (equation 4), using measured Chla, TP and SDD data (Carlson, 1977; Aizaki et al., 1981). The traditional TSI method used numbers (0-100) to express the state of a lake: TSI

Fig, 1S Correlation between Kd(PAR) and water transparency (SDD)

Fig. 2S Correlation between TSI and water transparency (SDD)

Holmes, R. W.: The secchi disk in turbid coastal water, Limnol. Oceanogr., 15(5), 688-694, 1970.

Ma, J., Song, K., Wen, Z., Zhao, Y., Shang, Y., Fang, C., Du, J.: Spatial Distribution of Diffuse Attenuation of Photosynthetic Active Radiation and Its Main Regulating Factors in Inland Waters of Northeast China, Remote Sensing, 8(11), 2016.

Raymont, J.E.G. Plankton and Productivity in the Oceans; Pergamon Press: Oxford, UK, 1967.

Aertebjerg, G.; Bresta, A.M. Guidelines for the Measurement of Phytoplankton Primary Production,

2nd ed.; Baltic Marine Biologists Publication: Charlottenlund, Denmark, 1984.

38. L 254-255: Sentence needs review.

Response: We have rewritten this sentence.

39. L 257-258: Stick to the abbreviations provided. It is confusing when alternating between calling the regions by name and abbreviation. '

Response: Thank you for the suggestion, and now we have used abbreviations throughout the manuscript.

40. L 260-264: Confusing and not very informative. Most of the readers have no idea about the locations of the referred lakes. Could be removed from the text. Perhaps authors may want to devote a bit more of effort to describe lakes that have high social-economic impact in the country, instead. Response: Thank you for the suggestion, and we have deleted the text. This part was rewritten.

41. L 269: "at all sampling sites" – what do you mean by that? Was the correlation tested for each site individually? It is not clear.

Response: We are very sorry for the confusing expression. The correlation was established based on the data in all sampling points.

42. L 271: What that the best linear regression, or was a linear regression the best one to describe the correlation? Have you tested for other types of functions?

Response: We also tested for other types of functions, such as exponential equation, logarithmic equation, and power equation, et al. The results showed that the best function to describe the relationship is a linear model.

43. L 279-280: "all the optically active components had impact on Kd(PAR)" – what do you mean by that?

Response: This means that three OACs: phytoplankton, CDOM, and inorganic particles all had influence on  $K_d(PAR)$ . The attenuation of photosynthetic active radiation in water was influenced by OACs, and the related results have been showed in Fig. 7.

44. L 282: what is the standardized coefficient of independent variables? Where are those results presented/discussed in the MS?

Response: The standardized coefficient is gotten from the multiple regression analysis. We listed it in Table 1.

| Table 1   | Summary | of multi | nle regress | sion anal | lvsis |
|-----------|---------|----------|-------------|-----------|-------|
| I doite I | Summary | or munu  |             | non ana   | 9313  |

|                                     | Standardized |                  |           |                         |            |       |
|-------------------------------------|--------------|------------------|-----------|-------------------------|------------|-------|
|                                     | coefficients |                  |           | Adjusted R 2 | Std. Error | Sig.  |
|                                     | асром        | a NAP | $a_{phy}$ |                         |            |       |
| All lakes                           | 0.130*       | 0.802*           | 0.217*    | 0.866                   | 0.833      | 0.000 |
| TSM <3.8 mg/L                       | 0.459*       | 0.408*           | 0.110*    | 0.742                   | 0.220      | 0.000 |
| TSM >3.8 mg/L (Non-eutrophic lakes) | 0.536*       | 0.381*           | -         | 0.770                   | 0.429      | 0.000 |
| TSM >3.8 mg/L (Eutrophic lakes)     | 0.086*       | 0.860*           | 0.210*    | 0.762                   | 1.106      | 0.000 |

Dependent Variable: Kd(PAR); Predictors: constant, aphy, aNAP, aCDOM; "\*" represents *p*<0.005.

45. L 285: The correlation between Kd and TSM was greater than for kd vs aNAP, how do the authors explain that, since they say that aNAP has the most significant impact on kd(PAR)?

Response: Among  $a_{NAP}$ ,  $a_{CDOM}$ , and  $a_{phy}$ ,  $a_{NAP}$  was the most important influence factor on Kd(PAR), followed by  $a_{phy}$ . In this study, the analysis and discussion was only confined to the light absorption characteristic of OACs. TSM was the total concentration of inorganic matters and phytoplankton in water. TSM is not just had the light absorption characteristic, but they could scatter the light. The scattering characteristics of TSM to light may have contribute to the better correlation between Kd and TSM than for Kd vs  $a_{NAP}$ .

46. L 294-295: Any reason/hypothesis why that region had the best performance for predicting kd(PAR)?

Response: TQR had the best performance for predicting  $K_d(PAR)$ , this may be related the natural environment and lake water characteristic, and we analyzed these factors in detail in part 4.1. The following factors, including the strong ultraviolet radiation, deep water, high salinity of water, low nutrient input in lake, and high water transparency in TQR, would result to the low aOACs and a good constituent stability of OACs. Thus the correlation between  $K_d(PAR)$  and  $a_{OACs}$  in TQR had the best fitting degree ( $R^2 = 0.85$ ) and the greatest relationship coefficient (slope=0.95) than in other limnetic regions.

47. L 303: those relative contributions are related to what?

Response: In all limnetic regions in this study,  $K_d(PAR)$  was dominated by inorganic particles absorption/scattering with mean relative contributions of 57.95%, followed by phytoplankton with mean relative contributions of 28.20%.

48. L 304-308: confusing sentences. The authors may want to rephrase them. L 328-329: Confusing sentence.

Response: Thank you for the comment. We have rephrased these sentences.

49. L 315: How was the 3.8 mg/L threshold defined? What is the reason for that value? L 320-322: What about the limnetic regions? Why to use this classification? Any reference for that? Is there any clustering for those stations? In figure 8 the TSM threshold division is further subdivided into non-eutrophy and eutrophy. Authors should state the reason for performing such divisions.

Response: Thank you for the comment. The TSM concentration of 3.8 mg/L used as a threshold to categorize the lakes is the result of regression tree analysis. We have added the related references to this analysis method in the manuscript (Breiman et al., 1984; Hampton et al., 2017). The result of this analysis showed that the tree had two branches with the boundary of 3.8 mg/L TSM. So the TSM concentration of 3.8 mg/L was used as a threshold to categorize the lakes in the subsequent analysis. Mean  $K_d(PAR)$  and standard error of  $K_d(PAR)$  were calculated for each branch of the regression trees. In figure 8 the TSM threshold division is further subdivided into non-eutrophy and eutrophy. That is because that the trophic status also had important effect on  $K_d(PAR)$  and SDD. We have analyzed the correlations between trophic state index (TSI) and water transparency (SDD), between water transparency (SDD) and  $K_d(PAR)$  to support this explain (Fig. 1S and 2S in comment 37).

Breiman L, Friedman J, Olshen R (1984) Classification and Regression Trees. Wadsworth International Group, Belmont

Hampton SE et al. (2017) Ecology under lake ice. Ecol Lett 20: 98-111.

50. L 318-319: why to combine oligo- and mesotrophic lakes?

Response: This is a good question. That is because either in oligotrophic lakes (TSM > 3.8 mg/L) or in mesotrophic lakes (TSM > 3.8 mg/L), the total number of sampling points was less than in eutrophic lakes. We try to make have the comparable sampling sites in different trophic status lakes. so we combine oligo- and mesotrophic lakes, this also can be called non-eutrophic lakes, then make the comparation between non-eutrophic lakes and eutrophic lakes.

51. L 336: what do you mean by "relational expression"?

Response: We are sorry for the ambiguity expression, and we have rephrased the sentence "the regression function was  $K_d(PAR)=0.30+0.48 \times a_{CDOM}+0.72 \times a_{NAP}+0.20 \times a_{phy}$  (R2=0.74, p<0.001)".

52. L 338-339: where are those results shown? L 339-340: why was aphy excluded? Authors have to estate the reason for that. Results – suggestion: I have the impression that the authors wanted to include all their results in the MS. However, some of those results could be omitted without changing the concept of the manuscript and it would make it easier to present, write and follow. Authors should rethink what results are worth it to be presented in the manuscript.

Response: Thank you for the suggestion. We have added the multiple regression analyses results were all showed in Table 1 (showed in Comment 44).

53. L 349-351: Based on what could you infer that? Are there any thresholds? Please, provide reference.

Response: In the present study, 47.37% of the in situ  $K_d(PAR)$  values ranged from 0.11 m-1 to 1.00 m-1, and 43.61% of  $K_d(PAR)$  ranged from 1.00 m-1 to 5.00 m-1. Based on the relationship between  $K_d(PAR)$  and water transparency (SDD) (Fig. 1S, see in Comment 37), 43.61% of sampling points had the transparency of 0.26-1.32 m, reflecting that approximately half of these sampling points are the turbid water body. Besides, the previous study has pointed out that when the lakes were in the eutrophic status, the SDD was lower than 1m, presenting by the turbid water (Olmanson, et al., 2008).

Olmanson, L. G., Bauer, M. E., Brezonik, P. L.: A 20-year Landsat water clarity census of Minnesota's 10,000 lakes, Remote Sens. Environ., 112(11), 4086-4097, 2008.

Carlson, R. E.: A trophic state index for lakes, Limnol. Oceanogr., 22(2), 361-369, 1977.

54. L 357-359: Please provide a reference for that statement.

- Response: We have added the references.
- Budzynska, A., Rosinska, J., Pelechata, A., et al.: Environmental factors driving the occurrence of the invasive cyanobacterium Sphaerospermopsis aphanizomenoides (Nostocales) in temperate lakes, Sci. Total Environ., 650, 1338-1347, 2019.
- Richardson, J., Miller, C., Maberly, S. C., et al.: Effects of multiple stressors on cyanobacteria abundance vary with lake type, Global Change Biol., 24(11), 5044-5055, 2018.
- 55. L 361-363: please provide reference.
- Response: We have added the reference.
- Ma, P. F., Wang, C. S., Meng, J., Ma, C., Zhao, X. X., Li, Y. L., Wang, M.: Late Oligocene-early Miocene evolution of the Lunpola Basin, central Tibetan Plateau, evidences from successive lacustrine records, Gondwana Res., 48, 224-236, 2017.
- 56. L 363-365: please provide reference.
- Response: We have added the reference.
- Yan, L., Sun, M., Yao, X., Gong, N., Li, X., Qi, M.: Lake water in the Tibet Plateau: Quality change and current status evaluation, Acta Scientiae Circumstantiae, 38(3), 900-910, 2018.
- 57. L 371-372: Do you mean wind-induced waves? What about the establishment of the seasonal

thermocline? What about the allochthnous input of TSM? What can you say about it?

Response: Thank you for the question. In this part, we discussed that the sediment re-suspension driven by wave disturbance would increase the TSM concentration in the shallow lakes. This sediment re-suspension occurred in shallow lakes, this shallow lakes did not have the seasonal thermocline phenomenon. In addition, the allochthnous input is a very important soure of TSM in the lake water, it is well known that the allochthnous input is ubiquity in all lakes. In this part, we tried to explain the reason of the higher TSM in NER than other area, so we emphasized the sediment re-suspension in shallow lakes.

58. L 379-381: The sentence does not add information to the discussion. Consider removing it or developing it more in deep.

Response: We have deleted it, thank you for the suggestion.

59. L 386: where are those similar inland bodies? Please develop more the discussion instead of making comparisons.

Response: Thank you for the question, we have listed these inland water bodies

60. L 401-404: CDOM photobleaching and photodegradation: how can the authors infer that based on their results? If the information is from other study, please, include the reference.

Response: We have added the reference in the revised manuscript (Song et al., 2018).

Song, K., Li, S., Wen, Z., Lyu, L., Shang, Y.: Characterization of chromophoric dissolved organic matter in lakes on the Tibet Plateau, China, using spectroscopic analysis, Biogeosciences Discuss., 2018, 1-50, 2018.

61. L 410-412: How can the authors infer that form their results? Have you measured phytoplankton biomass prior to the "overbloom"?

Response: This is very good questions, and we have added the references to support the conclusion, and some results have showed in Fig. 2.

- Duan, H., Ma, R., Xu, X., Kong, F., Zhang, S., Kong, W., Hao, J., Shang, L.: Two-Decade Reconstruction of Algal Blooms in China's Lake Taihu, Environmental Science & Technology, 43(10), 3522-3528, 2009.
- Zhao, H., Zhu, L., Wu, C., Meng, B., Zhou, Y., Jia, X.: Distribution characteristics analysis of algal bloom in Chaohu Lake based on the sky~earth collaborative method, China Environ. Sci., 38(6), 2297-2303, 2018.
- Shan, K., Li, L., Wang, X., Wu, Y., Hu, L., Yu, G., Song, L.: Modelling ecosystem structure and trophic interactions in a typical cyanobacterial bloom-dominated shallow Lake Dianchi, China, Ecol. Model., 291, 82-95, 2014.
- 62. L 412-415: Not clear what the authors want to say in that sentence.
- Response: We have deleted the references.
- 63. L 415-416: Many studies and only one reference?
- Response: We have added the references.
- Ma, J., Song, K., Wen, Z., Zhao, Y., Shang, Y., Fang, C., Du, J.: Spatial Distribution of Diffuse Attenuation of Photosynthetic Active Radiation and Its Main Regulating Factors in Inland Waters of Northeast China, Remote Sensing, 8(11), 2016.

- Zhang, Y. L., Zhang, B., Ma, R. H., Feng, S., Le, C. F.: Optically active substances and their contributions to the underwater light climate in Lake Taihu, a large shallow lake in China, Fundamental and Applied Limnology, 170(1), 11-19, 2007.
- 64. L 425: Awkward phrasing.

Response: We have rephrased the sentence.

65. L 428: Figure 5 only shows TSM results.

Response: We are sorry for the clerical error, this should be Figure 6.

66. L 428-429: What was the overall absorption budget for the studies regions? I think that your results would be clearer if you present a ternary plot for the absorption of OACs (CDOM x phytoplankton x NAP) for each of the regions.

Response: The overall absorption of OACs could explain 85% of  $K_d(PAR)$  in the whole studied region (Figure 4a). As you mentioned, the ternary plot for the absorption of OACs in each of the regions were added in the revised manuscript. This may be helpful in understand the absorption budget in studied regions.

---

## Author Comment (AC4) · 7 Dec 2018

**Response to reviewer**

**Title: Interactive comment on "Spatial pattern of $K_d$(PAR) and its relationship with light absorption of optically active components in inland waters across China**

**Referee: Rafael Gonçalves-Araujo**

19. L 135: Oligotrophic lakes: you use this terminology here and another few times along the MS and then changes to non-eutrophic lakes. It is necessary to have consistency when it comes to terminology.

Response: Thank you for the suggestion, and we would keep the consistency in the revised manuscript. We have used non-eutrophic lakes in the revised manuscript.

20. L 135-136: this sentence does not make sense. Consider rephrasing it.

Response: We have rephrased the sentence.

---

## Author Comment (AC5) · 7 Dec 2018

The study rationale and objectives are well stated and grounded in existing literature. Methodology is sound and adequately described, and conclusions of the study are supported by the data presented. The manuscript is publishable, but the text requires a great deal of editing. My detailed comments are listed below.

Response: We thank the reviewer for the very instructive and helpful suggestions for revision. We have revised manuscript according to the reviewer's suggestion, and the details are listed as following. Thank you very much again for the positive evaluation and giving us the chance to further improve our manuscript.

1. The abstract doesn't include all important from the paper: for example, the aim of the study was not even mentioned. The abstract should be more concrete too! "This study highlights the……"; "……findings which have application for monitoring Kd(PAR)..."

Response: Thank you for the suggestion, and we have rephrased the abstract as suggested (Page 2, line 17-34).

2. In manuscript, why are you going from OACs to $a_{OAC}$? Sometimes, you write OAC,

other time you write $a_{OACs}$. Unless there is a valid reason, it is important to use the same abbreviation throughout the manuscript.

Response: We are very sorry for the inconsistent expression, and we have checked the abbreviations throughout the manuscript. OACs is the abbreviation of optically active components, and $a_{OACs}$ represents the light absorption coefficient of optically active components. These are two different concepts. In our study, we analyzed the relationships between $a_{OACs}$ and $K_d(PAR)$.

3. Line 38 and throughout the manuscript, the citations should be presented as the followed format (First author, et al., year), please do not list all authors in the text.

Response: We completely agree with you, and we have revised them in the manuscript according to your suggestion.

4. Line 43 "optically active compounds" and "optically active components". You need to develop some consistency regarding the use of terms.

Response: Thank you for the suggestion, and we have consistently used "optically active components" in the revised manuscript.

5. Line 61-64 A reference is required here.

Response: Thank you for the suggestion, and we have added the reference "Prieur & Sathyendranath, 1981)" in the revised manuscript (Page 3, line 60).

Prieur, L., Sathyendranath, S.: An optical classification of coastal and oceanic waters based on the specific spectral absorption curves of phytoplankton pigments, dissolved organic matter, and other particulate materials, Limnol. Oceanogr., 26(4), 671-689, 1981.

6. Line 117-137 Condense these sentences. Limit your description to details that would help readers under the context of the study. This section is not your emphasis. You can reduce the length as soon as possible.

Line 138-144 Same thing here. These details are superfluous. Please do not in so much detail.

Response: We have accepted the suggestion, and have rephrased these sentences, thanks for the instructive comments (Page 7, line 106-152 ).

7. Line 160 I care how long the samples are stored in the field and how to be stored.

Response: The surface water was collected in the acid-washed HDPE bottles, and were placed in a portable refrigerator at 4 ℃ about 1-2 days before they were carried back to the laboratory.

8. Line 211 Has this classification been used before? If so, provide reference. In addition, and more importantly, this is the only time reference is made to CHAID approach in the entire manuscript. There is no reason to list the method, if it is not used in the analysis of data presented in the manuscript.

Response: Thank you for the comment. We have added the related references in the manuscript (Breiman et al., 1984; Hampton et al., 2017) (Page 9, line 185). The result of this analysis showed that the tree had two branches with the boundary of 3.8 mg/L TSM. So the TSM concentration of 3.8 mg/L was used as a threshold to categorize the lakes in the subsequent analysis. Mean $K_d$(PAR) and standard error of $K_d$(PAR) were calculated for each branch of the regression trees.

Breiman L, Friedman J, Olshen R (1984) Classification and Regression Trees. Wadsworth International Group, Belmont

Hampton SE et al. (2017) Ecology under lake ice. Ecol Lett 20: 98-111.

9. Line 221 I think, it's more precise to say "point". An entire lake can be viewed as a study site.

Response: We completely agree with you, and we have used "point" in the manuscript.

10. Line 227 That sentence is a repetition of information that has just been provided.

Response: We have deleted the sentence as suggested.

11. Line 234 Sometimes you use trophic status but sometimes trophic state. Please the consistent expression for all the text.

Figure 2 and Figure 8: I could not find in the text how the trophic states were defined.

Response: We have consistently used "trophic status" in the revised manuscript, and added the definition of trophic status. The assessment of the trophic status of lakes was based on the modified Carlson's trophic state index (TSI), using measured *Chla*, TP and SDD data. The TSI value was calculated from TSI(Chla), TSI(SD) and TSI(TP), see Equations 4-7 (Carlson, 1977; Aizaki et al., 1981). The traditional TSI method used numbers (0-100) to express the state of a lake: TSI <30 indicates oligotrophic state, 30 - 50 indicates mesotrophic state, and 50 - 100 indicates eutrophic state. (Page 9, line 189-193).

$$TSI_M(Chla) = 10 \times (2.46 + \frac{\ln Chla}{\ln 2.5}) \tag{1}$$

$$TSI_M(SD) = 10 \times (2.46 + \frac{3.69 - 1.52 \times \ln SD}{\ln 2.5}) \tag{2}$$

$$TSI_M(TP) = 10 \times (2.46 + \frac{6.71 + 1.15 \times \ln(TP)}{\ln 2.5})$$

(3)

$$TSI = 0.54 \times TSI_M(Chla) + 0.297 \times TSI_M(SD) + 0.163 \times TSI_M(TP)$$

(4)

Aizaki, M., Otsuki, A., Fukushima, T., Kawai, T., Hosomi, M., Muraoka, K. 1981. Application of modified Carlson's trophic state index to Japanese lakes and its relationship to other parameters related to trophic state.

Carlson, R. E.: A trophic state index for lakes, Limnol. Oceanogr., 22(2), 361-369, 1977.

12. Line 314 What is the basis for using a TSM threshold of 3.8 mg/L to categorize the lakes? If that is based on a previous study, list the reference(s). If that is based on the analysis of the data from the present study, indicate where the information is presented.

**Response:** Thank you for the comment. The TSM concentration of 3.8 mg/L used as a threshold to categorize the lakes is the result of regression tree analysis. We have explained the method in the former comment. The authors really thank for the very instructive comments.